# Neural population dynamics in motor cortex are different for reach and grasp

**Aneesha K Suresh**[1†], **James M Goodman**[1†], **Elizaveta V Okorokova**[1], **Matthew Kaufman**[1,2,3], **Nicholas G Hatsopoulos**[1,2,3], **Sliman J Bensmaia**[1,2,3*]

[1]Committee on Computational Neuroscience, University of Chicago, Chicago, United States; [2]Department of Organismal Biology and Anatomy, University of Chicago, Chicago, United States; [3]Grossman Institute for Neuroscience, Quantitative Biology and Human Behavior, University of Chicago, Chicago, United States

**Abstract** Low-dimensional linear dynamics are observed in neuronal population activity in primary motor cortex (M1) when monkeys make reaching movements. This population-level behavior is consistent with a role for M1 as an autonomous pattern generator that drives muscles to give rise to movement. In the present study, we examine whether similar dynamics are also observed during grasping movements, which involve fundamentally different patterns of kinematics and muscle activations. Using a variety of analytical approaches, we show that M1 does not exhibit such dynamics during grasping movements. Rather, the grasp-related neuronal dynamics in M1 are similar to their counterparts in somatosensory cortex, whose activity is driven primarily by afferent inputs rather than by intrinsic dynamics. The basic structure of the neuronal activity underlying hand control is thus fundamentally different from that underlying arm control.

*For correspondence:
sliman@uchicago.edu

†These authors contributed
equally to this work

Competing interests: The
authors declare that no
competing interests exist.

Reviewing editor: Samantha R
Santacruz, The University of
Texas at Austin, United States

## Introduction

The responses of populations of neurons in primary motor cortex (M1) exhibit rotational dynamics – reflecting a neural oscillation at the population level – when animals make arm movements, including reaching and cycling (***Churchland et al., 2012***; ***Lara et al., 2018a***; ***Russo et al., 2018***; ***Shenoy et al., 2013***). One interpretation of this population-level behavior is that M1 acts as a pattern generator that drives muscles to give rise to movement. A major question is whether such population dynamics reflect a general principle of M1 function, or whether they underlie some behaviors and effectors but not others. To address this question, we examined the dynamics in the neuronal population activity during grasping movements, which involve a plant (the hand) that serves a different function, comprises more joints, and is characterized by different mechanical properties (***Rathelot and Strick, 2009***). While the hand is endowed with many degrees of freedom, hand kinematics can be largely accounted for within a small subspace (***Ingram et al., 2008***; ***Overduin et al., 2015***; ***Santello et al., 1998***; ***Tresch and Jarc, 2009***) so we might expect to observe low-dimensional neural dynamics during hand movements, not unlike those observed during arm movements.

To test this, we recorded the neural activity in M1 using chronically implanted electrode arrays as monkeys performed a grasping task, restricting our analyses to responses before object contact (***Figure 1—figure supplement 1***). Animals were required to hold their arms still at the elbow and shoulder joints as a robotic arm presented each object to their contralateral hand. This task – which can be likened to catching a tossed object or grasping an offered one – limits proximal limb movements and isolates grasping movements. For comparison, we also examined the responses of M1 neurons during a center-out reaching task (***Hatsopoulos et al., 2007***). In addition, we compared grasping responses in M1 to their counterparts in somatosensory cortex (SCx), which is primarily driven by afferent input and therefore should not exhibit autonomous dynamics (***Russo et al., 2018***).

## Results

First, we used jPCA to search for rotational dynamics in a low-dimensional manifold of M1 population activity (*Figure 1*; *Churchland et al., 2012*). Replicating previous findings, reaching was associated with a variety of different activity patterns at the single-neuron level (*Figure 1A*) that were collectively governed by rotational dynamics at the population level (*Figure 1C,E*). During grasp, individual M1 neurons similarly exhibited a variety of different response profiles (*Figure 1B*), but rotational dynamics were weak or absent at the population level (*Figure 1D,E*).

Given the poor fit of rotational dynamics to neural activity during grasp, we next assessed whether activity could be described by a linear dynamical system of any kind. To test for linear dynamics, we fit a regression model using the first 10 principal components of the M1 population activity (x(t)) to predict their rates of change (dx/dt). We found x(t) to be far less predictive of dx/dt in grasp than in reach, suggesting much weaker linear dynamics in M1 during grasp (*Figure 1F*). We verified that these results were not an artifact of data alignment, movement epoch, peak firing rate, smoothing, population size, or number of behavioral conditions (*Figure 1—figure supplement 2*).

The possibility remains that dynamics are present in M1 during grasp, but that they are higher-dimensional or more nonlinear than during reach. Indeed, M1 population activity during a reach-grasp-manipulate task is higher dimensional than is M1 activity during reach alone (*Rouse and Schieber, 2018*). In light of this, we used Latent Factor Analysis via Dynamical Systems (LFADS) to infer and exploit latent dynamics and thereby improve estimation of single-trial firing rates, then applied a decoder to evaluate the level of improvement. Naturally, the benefit of LFADS is only realized if the neural population acts like a dynamical system. Importantly, such dynamics are minimally constrained and can, in principle, be arbitrarily high dimensional and/or highly nonlinear. First, as expected, we found that in both datasets, neural reconstruction of single trials improved with LFADS (*Figure 2—figure supplement 1A,B*). However, LFADS yielded a significantly greater improvement in reconstruction accuracy for reach than for grasp (t(311) = 7.07, p=5.11e-12; *Figure 2—figure supplement 1B*). Second, a standard Kalman filter was used to decode joint angle kinematics from the inferred latent factors (*Figure 2*). If latent dynamics in M1 play a key role in the generation of temporal sequences of muscle activations, which in turn give rise to movement, LFADS should substantially improve kinematic decoding. Replicating previous results, we found decoding accuracy to be substantially improved for reaching when processing firing rates using LFADS (*Figure 2A,C*) ($R^2$ = 0.93 and 0.57 with and without LFADS, respectively). In contrast, LFADS offered minimal improvement in accuracy when decoding grasping kinematics in two monkeys (*Figure 2B,C*) ($R^2$ = 0.46 and 0.37), regardless of the latent dimensionality of the model (*Figure 2—figure supplement 1C*) or whether external inputs were included (*Figure 2—figure supplement 1D*). These decoding results demonstrate that the strong dynamical structure seen in the M1 population activity during reach is not observed during grasp, even when dimensionality and linearity constraints are relaxed.

As a separate way to examine the neural dynamics in grasping responses, we computed a neural 'tangling' metric, which assesses the degree to which network dynamics are governed by a smooth and consistent flow field (*Russo et al., 2018*). In a smooth, autonomous dynamical system, neural trajectories passing through nearby points in state space should have similar derivatives. The tangling metric (Q) assesses the degree to which this is the case over a specified (reduced) number of dimensions. During reaching, muscle activity and movement kinematics have been shown to exhibit more tangling than does M1 activity, presumably because the cortical circuits act as a dynamical pattern generator whereas muscles are input-driven (*Russo et al., 2018*). We replicated these results for reaching: neural activity was much less tangled than the corresponding arm kinematics (position, velocity, and acceleration of joint angles) (*Figure 3A*), as long as the subspace was large enough (>2D), (*Figure 3—figure supplement 1*). For grasp, however, M1 activity was as tangled as the corresponding hand kinematics, or even more so (*Figure 3B*), over all subspaces (*Figure 3—figure supplement 1*). Next, we compared tangling in the grasp-related activity in M1 to its counterpart in SCx, which, as a sensory area, is expected to exhibit tangled activity (as shown during reaching movements [*Russo et al., 2018*]). Surprisingly, population activity patterns in both M1 and SCx were similarly tangled during grasp (*Figure 3C*). In summary, M1 responses during grasp do not exhibit the properties of an autonomous dynamical system, but rather are tangled to a similar degree as are sensory cortical responses (*Figure 3D*).

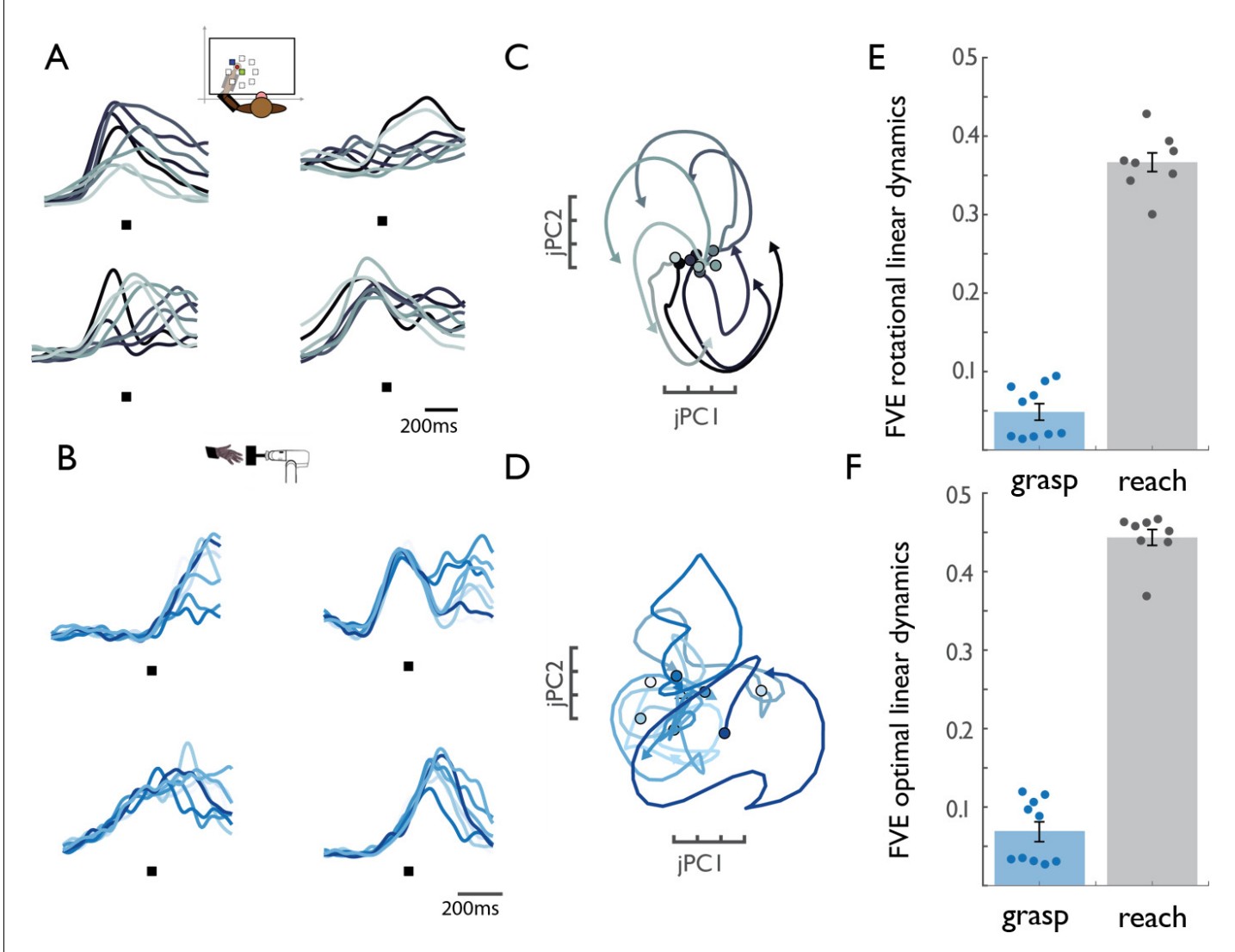

**Figure 1.** M1 rotational dynamics during reaching and grasping. (A) Normalized peri-event histograms aligned to movement onset (black square) for four representative neurons during the reaching task (Monkey 4, Dataset 5). Each shade of gray indicates a different reach direction, trial-averaged for each reaching condition (eight total). (B) Normalized peri-event histograms aligned to maximum aperture (black square) for four representative neurons during the grasping task (Monkey 2, Dataset 2). Each shade of blue indicates a neuron's response, trial-averaged for different object groups. (C) Rotational dynamics in the population response during reaching for Monkey 4 (Dataset 5) projected onto the first jPCA plane. Different shades of gray denote different reach directions. (D) Lack of similar M1 rotational dynamics during grasping. Different shades of blue indicate different object groups, for Monkey 2 (Dataset 2). (E) FVE (fraction of variance explained) in the rate of change of neural PCs (dx/dt) explained by the best fitting rotational dynamical system. The difference in FVE for reach and grasp is significant (two-sample two-sided equal-variance t-test, t(16) = −19.44, p=4.67e-13). Error bars denote standard error of the mean and data points represent the outcomes of cross-validation folds (across conditions – see Materials and methods) for each of two monkeys. (F) FVE in the rate of change of neural PCs (dx/dt) explained by the best fitting linear dynamical system, not constrained to be rotational. The difference in FVE is highly significant (two-sample two-sided equal-variance t-test, t(16) = −21.37 p=1.57e-14). Error bars denote standard error of the mean and data points represent the outcomes of cross-validation folds for each of two monkeys (fourfold for reaching data, and 5-fold for grasping data). The lack of dynamical structure during grasping relative to reach is further established in a series of control analyses (*Figure 1—figure supplement 1*).

The online version of this article includes the following figure supplement(s) for figure 1:

**Figure supplement 1.** Grasping behavior and neurophysiology.

**Figure supplement 2.** Control analyses for reaching and grasping.

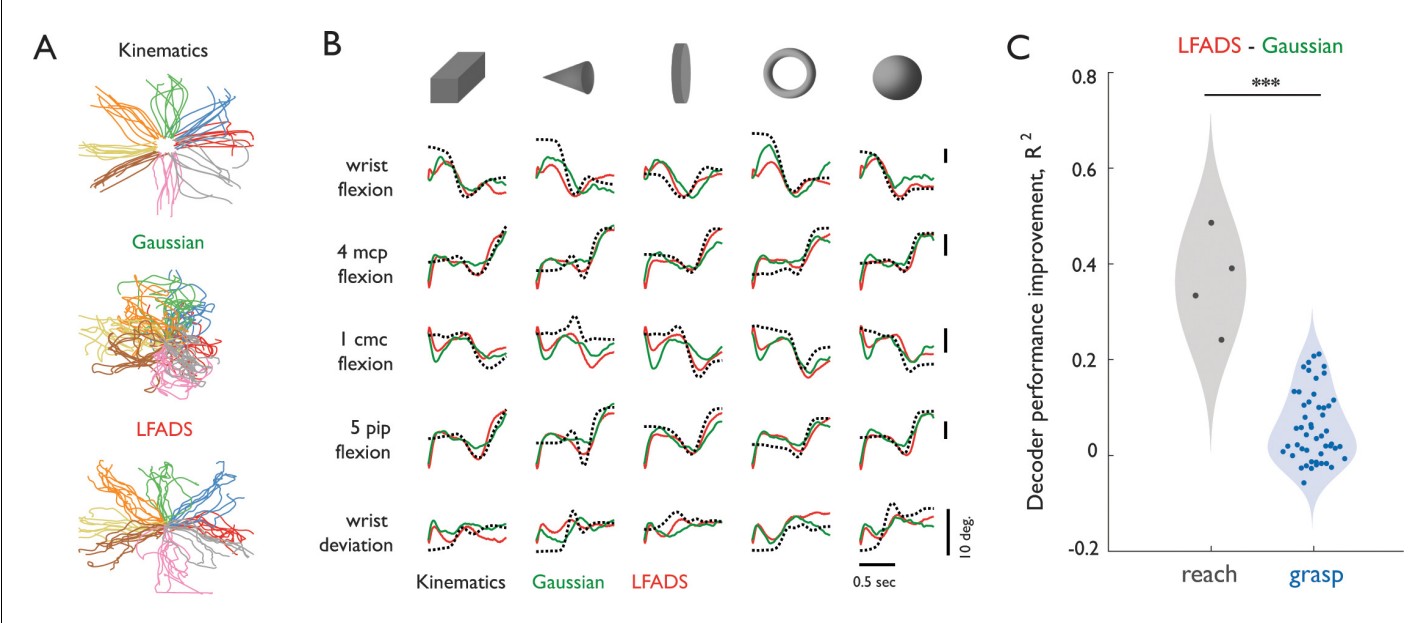

**Figure 2.** Decoding of kinematics based on population activity pre-processed with Gaussian smoothing or with LFADS. (**A**) End-point coordinates of center-out reaching with actual kinematics (top) or kinematics reconstructed with neural data preprocessed with Gaussian smoothing (middle) or LFADS (bottom). Coordinates are color-coded according to the eight directions of movement. While conditions are visually separable in both Gaussian and LFADS reconstructions, the later provides a smoother and more reliable estimate. (**B**) Single-trial time-varying angles of five hand joints (black, dashed) from monkey three as it grasped five objects along with their decoded counterparts (Gaussian-smoothed in green, LFADS-inferred in red). Both Gaussian-smoothed and LFADS-inferred firing rates yield similar decoding errors. Here, '4mcp flexion' refers to flexion/extension of the fourth metacarpophalangeal joint; '5pip flexion' - flexion/extension of the fifth proximal interphalangeal joint; and '1cmc flexion' - flexion/extension of the first carpo-metacarpal joint. (**C**) Difference in performance gauged by the coefficient of determination between decoders with LFADS and Gaussian smoothing for reach (gray) and grasp (blue). Each point denotes the mean performance increase across 10-fold cross-validation of all degrees of freedom pooled across monkeys for reach (2 monkeys with 2 DoFs each) and grasp (2 monkeys with 22 and 29 DoFs, respectively). All decoders were fit using a population of 37 M1 neurons. LFADS leads to significantly larger decoder performance improvement for reach than for grasp. Stars indicate significance of a Mann-Whitney-Wilcoxon test for unmatched samples: *** - alpha of 0.001 for one-sided alternative hypothesis.

The online version of this article includes the following figure supplement(s) for figure 2:

**Figure supplement 1.** Validation of LFADS.

## Discussion

We find that M1 does not exhibit low-dimensional dynamics during grasp as it does during reach (*Churchland et al., 2012*), reach-to-grasp (*Rouse and Schieber, 2018*), or reach-like center-out pointing (*Pandarinath et al., 2015*). The difference between reach- and grasp-related neuronal dynamics seems to stem from the fundamentally different kinematics and functions of these movements, rather than from effector-specific differences, since dynamics are observed for reach-like finger movements. That rotational dynamics are observed in reach-to-grasp likely reflects the reaching component of the behavior, consistent with the observation that movement signals are broadcast widely throughout motor cortex (*Musall et al., 2019*; *Stavisky et al., 2019*; *Willett et al., 2020*).

Other factors might also explain the different dynamical profiles in M1 between reach and grasp. One might conjecture that M1 population dynamics are much higher dimensional and/or more non-linear for grasp than for reach, which might explain our failure to detect dynamics in grasp-related M1 activity. However, both LFADS (*Pandarinath et al., 2018*; *Figure 2—figure supplement 1*) and the tangling metric (*Figure 3—figure supplement 1*) can accommodate high-dimensional systems and some degree of nonlinearity in the dynamics. We verified that our failure to observe dynamics did not stem from a failure to adequately characterize a high-dimensional grasp-related response in M1 commensurate with the dimensionality of the movement (See 'Dimensionality of the neuronal response' in the Materials and methods, *Figure 3—figure supplement 2*). We cannot exclude the possibility that dynamics may be observed in a much higher dimensional space than we can resolve

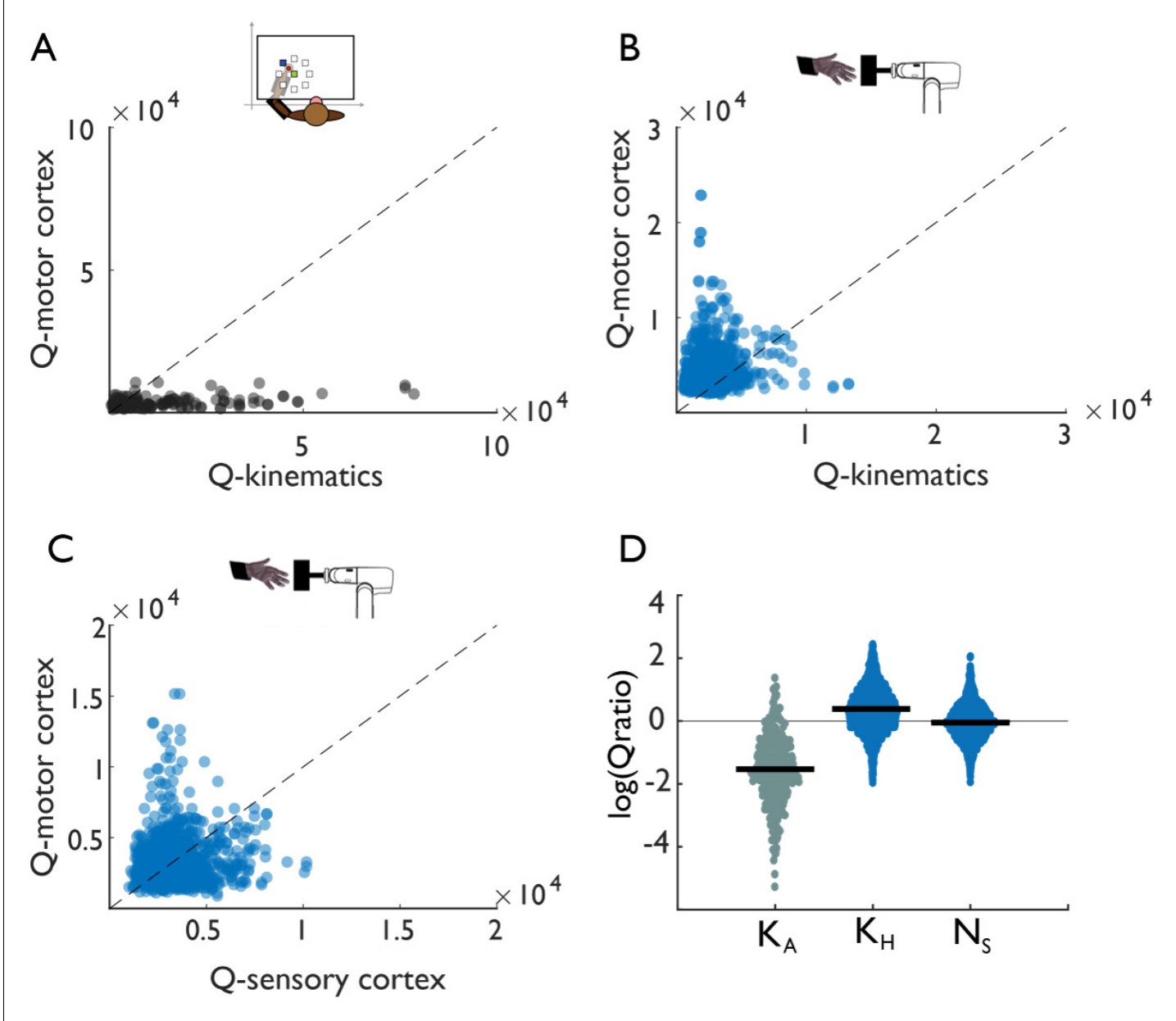

**Figure 3.** Tangling in reach and grasp. (**A**) Tangling metric (Q) for population responses in motor cortex vs. Q for kinematics during reaching. Kinematic tangling is higher than neural tangling, consistent with motor cortex acting as a pattern generation during reach. (**B**) Q-M1 population vs. Q-kinematics for grasping. Neural tangling is higher than kinematic tangling, which argues against pattern generation as the dominant mode during grasp. (**C**) Q-M1 population vs. Q-SCx population. Neural tangling is similar in M1 and SCx. For plots A-C, each point represents the max Q value for a (trial-averaged) neural state at a single time point and single task condition for one monkey (Monkey 1, Dataset 1). (**D**) Log of Q-motor/Q-kinematics of the arm during reach ($K_A$), Q-motor/Q-kinematics of the hand during grasp ($K_H$), and Q-motor/Q-sensory during grasp ($N_s$). Each point represents the log-ratio for a single condition and time point (pooled across two monkeys each). Black bars denote the mean log-ratio. The differences between reaching-derived and grasping-derived log-ratios are significant and substantial (two-sample two-sided equal-variance t-test: $K_H$ | $t(2978)=-43$, p=1.03e-130; $N_s$ |$t(2978)=-39$ p=1.87e-121). Tangling is insensitive to the precise dimensionality, provided it exceeds a minimum dimensionality (*Figure 3— figure supplement 1*).

The online version of this article includes the following figure supplement(s) for figure 3:

**Figure supplement 1.** Tangling vs. dimensionality.

**Figure supplement 2.** Dimensionality of grasp-related neuronal responses.

with our sample, one whose dimensionality far exceeds that of the movement itself. To test this hypothesis will require large-scale neural recordings obtained during grasping.

Another possibility is that M1 dynamics are under greater influence from extrinsic inputs for grasp than for reach: inputs can push neuronal activity away from the trajectories dictated by the intrinsic dynamics, thereby giving rise to tangling. M1 receives input from large swaths of the brain that each exhibit their own dynamics, including the supplementary motor area (*Lara et al., 2018b*; *Russo et al., 2020*), premotor and posterior parietal cortices (*Michaels et al., 2018*), and motor thalamus (*Sauerbrei et al., 2020*), in addition to responding to somatosensory and visual inputs (*Suminski et al., 2010*). Our findings are consistent with the hypothesis that grasp involves more inputs to M1 than does reach, or that grasp-related inputs are more disruptive to the intrinsic dynamics in M1 than are their reach-related counterparts (*Figure 2—figure supplement 1*).

Whatever the case may be, the low-dimensional linear dynamics observed in M1 during reaching are not present during grasping, consistent with an emerging view that the cortical circuits that track and control the hand differ from those that track and control the proximal limb (*Goodman et al., 2019*; *Rathelot and Strick, 2009*).

## Materials and methods

### Behavior and neurophysiological recordings for grasping task

We recorded single-unit responses in the primary motor and somatosensory cortices (M1 and SCx) of two monkeys (Macaca mulatta) (M1: $N_1 = 58$, $N_2 = 53$ | SCx: $N_1 = 26$ $N_2 = 28$) as they grasped each of 35 objects an average of 10 times per session. We refer to these recordings as Dataset 1 and Dataset 2, which were recorded from Monkey 1 and Monkey 2, respectively. Neuronal recordings were obtained across 6 and 9 sessions, respectively, and are used in the jPCA and tangling analyses. We also recorded simultaneously from populations of neurons in M1 in two monkeys ($N_3 = 44$, $N_4 = 37$) during a single session of this same task. These are called, respectively, Dataset 3 and Dataset 4. The first of these ($N_3$) was recorded from a third Monkey, Monkey 3; the second population of simultaneously recorded neurons ($N_4$) was obtained from the same monkey (Monkey 1) as the first set of sequentially recorded neurons ($N_1$). The recordings in Monkey 1 were achieved with different arrays and separated by 3 years. Simultaneously recorded populations were used for the decoding analyses.

On each trial (*Figure 1—figure supplement 1*), one of 25 objects was manually placed on the end of an industrial robotic arm (MELFA RV-1A, Mitsubishi Electric, Tokyo, Japan). After a 1–3 s delay, randomly drawn on a trial-by-trial basis, the robot translated the object toward the animal's stationary hand. The animal was required to maintain its arms in the primate chair for the trial to proceed: if light sensors on the arm rest became unobstructed before the robot began to move, the trial was aborted. We also confirmed that the animal produced minimal proximal limb movement by inspecting videos of the experiments and from the reconstructed kinematics. The object began 12.8 cm from the animal's hand and followed a linear trajectory toward the hand at a constant speed of 16 cm/s for a duration of 800 ms. As the object approached, the animal shaped its hand to grasp it. Some of the shapes were presented at different orientations, requiring a different grasping strategy, yielding 35 unique 'objects'. Each object was presented 8–11 times in a given session.

The timing of start of movement, maximum aperture, and grasp events were inferred on the basis of the recorded kinematics. A subset of trials from each session were manually scored for each of these three events. On the basis of these training data, joint angular kinematic trajectories spanning 200 ms before and after each frame were used as features to train a multi-class linear discriminant classifier to discriminate among these four classes: all three events of interest and 'no event'. Log likelihood ratio was used to determine which 'start of movement', 'maximum aperture', and 'grasp' times were most probable relative to 'no event'. Events were sequentially labeled for each trial to enforce the constraint that start of movement precedes maximum aperture, and maximum aperture precedes grasp. The median interval between the start of movement and maximum aperture was 450 ± 85 ms (median ± interquartile range) for Monkey 1 (across both sets of recordings), 240.0 ± 10.0 ms for Monkey 2, and 456 ± 216 ms for Monkey 3. The interval between maximum aperture and grasp was 356 ± 230 ms for Monkey 1, 410 ± 160 ms for Monkey 2, and 274 ± 145 ms

for Monkey 3. Total grasp times from start of movement to grasp were 825 ± 280 ms for Monkey 1, 650 ± 170 ms for Monkey 2, and 755 ± 303 ms for Monkey 3.

Neural recordings were obtained from two monkeys ($N_1$ and $N_2$) using semi-chronic electrode arrays (SC96 arrays, Gray Matter Research, Bozeman, MT) (*Dotson et al., 2017*; *Figure 1—figure supplement 1*). Electrodes, which were individually depth-adjustable, were moved to different depths on different sessions to capture new units. Units spanning both M1 and SCx were recorded using these arrays, and SCx data comprise populations from both proprioceptive subdivisions of SCx, namely, Brodmann's areas 3a and 2. Simultaneous neural recordings were obtained from one monkey ($N_3$) using a combination of Utah electrode arrays (UEAs, Blackrock Microsystems, Inc, Salt Lake City, UT) and floating microelectrode arrays (FMAs, Microprobes for Life Science, Gaithersburg, MD) targeting rostral and caudal subdivisions of the hand representation of M1, respectively. In the other monkey ($N_4$), simultaneous population recordings were obtained using a single 64-channel Utah array targeting the hand representation of (rostral) M1. Single units from all sessions (treated as distinct units) were extracted using an Offline Sorter (Plexon Inc, Dallas TX). Units were identified based on inter-spike interval distribution and waveform shape and size.

Hand joint kinematics, namely the angles and angular velocities about all motile axes of rotation in the joints of the wrist and digits, were tracked at a rate of 100 Hz by means of a 14-camera motion tracking system (MX-T series, VICON, Los Angeles, CA). The VICON system tracked the three-dimensional positions of the markers, and time-varying joint angles were computed using inverse kinematics based on a musculoskeletal model of the human arm (https://simtk.org/projects/ulb_project) (*Anderson and Pandy, 2001*; *Anderson and Pandy, 1999*; *de Leva, 1996*; *Delp et al., 1990*; *Dempster and Gaughran, 1967*; *Holzbaur et al., 2005*; *Yamaguchi and Zajac, 1989*) implemented in Opensim (https://simtk.org/frs/index.php?group_id=91) (*Delp et al., 2007*) with segments scaled to the sizes of those in a monkey limb using Opensim's built-in scaling function. Task and kinematic recording methods are reported in an earlier publication (*Goodman et al., 2019*). We used a linear discriminant classifier as detailed in this previous publication to determine whether objects indeed evoked distinct kinematics (*Figure 1—figure supplement 1*).

All surgical, behavioral, and experimental procedures conformed to the guidelines of the National Institutes of Health and were approved by the University of Chicago Institutional Animal Care and Use Committee.

## Behavior and neurophysiological recordings for reaching task

To compare grasp to reach, we analyzed previously published single- and multi-unit responses from two additional M1 populations (M1: $N_5$ = 76, $N_6$ = 107) recorded from two additional monkeys (Monkeys 4 and 5, respectively) operantly trained to move a cursor in a variable delay center out reaching task (*Hatsopoulos et al., 2007*). These recordings are called Dataset 5 and Dataset 6, respectively. The monkey's arm rested on cushioned arm troughs secured to links of a two-joint exoskeletal robotic arm (KINARM system; BKIN Technologies, Kingston, Ontario, Canada) underneath a projection surface. The shoulder and elbow joint angles were sampled at 500 Hz by the motor encoders of the robotic arm, and the *x* and *y* positions of the hand were computed using the forward kinematic equations. The center-out task involved movements from a center target to one of eight peripherally positioned targets (5 to 7 cm away). Targets were radially defined, spanning a full 360° rotation about the central target in 45° increments. Each trial comprised two epochs: first, an instruction period lasting 1 to 1.5 s, during which the monkey held its hand over the center target to make the peripheral target appear; second, a 'go' period, cued by blinking of the peripheral target, which indicated to the monkey that it could begin to move toward the target. Following the 'go' cue, movement onset was 324 ± 106 ms (median ± interquartile range) for Monkey 4 in Dataset 5, and 580 ± 482 ms for Monkey 5 in Dataset 6. Total movement duration was 516 ± 336 ms for Monkey 4 in Dataset 5 and 736 ± 545 ms for Monkey 5 in Dataset 6. Single- and multi-unit activities were recorded from each monkey during the course of a single session using an UEA implanted in the upper limb representation of contralateral M1. All surgical, behavioral, and experimental procedures conformed to the guidelines of the National Institutes of Health and were approved by the University of Chicago Institutional Animal Care and Use Committee.

Information about all grasping and reaching datasets and their associated analyses is provided in Table 1 of *Supplementary file 1*.

## Differences between reach and grasp and their potential implications for population dynamics

In this section, we discuss differences between the reach and grasp tasks that might have had an impact on the neuronal dynamics.

First, movements were cued differently in the two tasks. For reaching, targets blinked to cue movement. For grasping, there was no explicit movement cue; rather, the animals could begin pre-shaping their hand as soon as the robot began to move, although they had to wait for the object to reach the hand to complete their grasp and obtain a reward. Nonetheless, we found that the delay between the beginning of the robot's approach and hand movement onset (median ± interquartile range: Monkey 1 – 271 ± 100 ms; Monkey 2 – 419 ± 101 ms; numbers not available for Monkey 3) was similar to the delay in the reaching task between the go cue and start of movement. Note, moreover, that the nature of the 'delay' period should have little effect on neuronal dynamics. Indeed, self-initiated reaches and those that are executed rapidly with little to no preparation are nonetheless associated with rotational M1 dynamics (*Lara et al., 2018a*).

Second, the kinematics of reaching and grasping are quite different, and differences in the respective ranges of motion or speeds could mediate the observed differences in neuronal dynamics. However, the ranges of motion and distribution of speeds were similar for reach and grasp (*Figure 1—figure supplement 1C–D,G*), suggesting that the observed differences in neuronal dynamics are not trivial consequences of differences in the kinematics. On a related note, grasping movements with no reach (lasting roughly 700 ms) were generally slower than those reported in in the context of reach (lasting roughly 300 ms) (*Bonini et al., 2014*; *Chen et al., 2009*; *Lehmann and Scherberger, 2013*; *Rouse and Schieber, 2015*; *Roy et al., 2000*; *Theverapperuma et al., 2006*), as the animals had to wait for the robot to transport the object to their hand. Note, however, that similar constraints on movement duration and speed during reaching do not affect the presence or nature of M1 rotational dynamics during those movements (*Churchland et al., 2012*). As such, speed differences should not lead to qualitatively different M1 population dynamics.

Third, we considered whether grasping without reaching might simply be too 'unnatural' to be controlled by stereotyped M1 dynamics. However, we observed the presence of two hallmarks of grasping behavior: a clearly-defined maximum aperture phase and the presence of hand pre-shaping (*Jeannerod, 1984*; *Jeannerod, 1981*; *Santello et al., 2002*; *Santello and Soechting, 1998*). The latter is evidenced by a gradual improvement in our ability to classify objects based on the kinematics they evoke as the trial proceeded (*Figure 1—figure supplement 1E*): Upon start of movement, the hand is in a generic configuration that is independent of the presented object. However, as the trial proceeds, hand kinematics become increasingly object-specific, culminating in a high classification performance just before object contact. Furthermore, grasping kinematics have been previously shown to be robust to different types of reaches (*Wang and Stelmach, 1998*).

## Data analysis

### Data pre-processing

For both reach and grasp, neuronal responses were aligned to the start of movement, resampled at 100 Hz so that they would be at the same temporal resolution, averaged across trials, then smoothed by convolution with a Gaussian (25 ms S.D.). For jPCA, we then followed the data pre-processing steps described in *Churchland et al., 2012*: normalization of individual neuronal firing rates, subtraction of the cross-condition mean peri-event time histogram (PETH) from each neuron's response in each condition, and applying principal component analysis (PCA) to reduce the dimensionality of the population response. For LFADS and the tangling analyses, only the normalization of neurons' firing rates was performed. Although the condition-invariant response varies in a meaningful way (*Kaufman et al., 2016*), its inclusion obstructs our ability to use jPCA to visualize neural trajectories whose initial conditions vary, and thus our ability to use jPCA to evaluate claims of dynamical structure. Even when this component is especially large, dynamical structure in the remaining condition-dependent neural activity has been observed (*Rouse and Schieber, 2018*), thus subtraction of even a large condition-independent response should permit the inference of neural dynamics. We used 10 dimensions instead of six (*Churchland et al., 2012*) as a compromise between the lower dimensional reach data and the higher dimensional grasp data.

## jPCA

We applied to the population data (reduced to 10 dimensions by PCA) a published dimensionality reduction method, jPCA (*Churchland et al., 2012*), which finds orthonormal basis projections that capture rotational structure in the data. We used a similar number of dimensions for both reach and grasp, as PCA revealed no stark differences in the effective dimensionality of the neural population between the two tasks (*Figure 1—figure supplement 1F*). With jPCA, the neural state is compared with its derivative, and the strictly rotational dynamical system that explains the largest fraction of variance in that derivative is identified. The delay periods between the presentation/go-cue for each monkey varied, along with the reaction times, so we selected a single time interval (averaging 700 ms) that maximized rotational variance across all of them. For the reach data, data were aligned to the start of movement and the analysis window was centered on this event, whereas for the grasp data, data were aligned to maximum hand aperture, and we analyzed the interval centered on this event. In some cases, the center of this 700 ms window was shifted between −350 ms to +350 ms relative to the alignment event to obtain an estimate of how rotational dynamics change over the course of the trial (e.g. *Figure 1—figure supplement 2*). These events were chosen for alignment as they were associated with both the largest peak firing rates and the strongest rotational dynamics. Other alignment events were also tested, to test robustness (*Figure 1—figure supplement 2B*).

## Object clustering

Each of the 35 objects was presented 10 times per session, which yields a smaller number of trials per condition than were used to assess jPCA during reaching (at least 40). To permit pooling across a larger number of trials when visualizing and quantifying population dynamics with jPCA (*Figure 1*), objects in the grasp task were grouped into eight object clusters on the basis of the trial-averaged similarity of hand posture across all 30 joint degrees of freedom 10 ms prior to grasp (i.e. object contact). Objects were hierarchically clustered into eight clusters on the basis of the Ward linkage function (MATLAB `clusterdata`). Eight clusters were chosen to match the number of conditions in the reaching task. Cluster sizes were not uniform; the smallest comprised two and the largest nine different objects, with the median cluster comprising four objects.

As the clustering method just described yielded different cluster sizes, we assessed an alternative clustering procedure (*Figure 1—figure supplement 2F*) that guaranteed objects were divided into seven equally-sized clusters (five objects per cluster). Rather than determining cluster membership on the basis of a linkage threshold, cluster linkages were instead used to sort the objects on the basis of their dendrogram placements (MATLAB `dendrogram`). Clusters were obtained by grouping the first five objects in this sorted list into a common cluster, then the next five, and so on. This resulted in slightly poorer performance of jPCA (see *Quantification*).

For completeness, we also assessed jPCA without clustering (*Figure 1—figure supplement 2E*), which also resulted in slightly poorer performance of jPCA and was considerably more difficult to visualize given the large number of conditions.

## Quantification

In a linear dynamical system, the derivative of the state is a linear function of the state. We wished to assess whether a linear dynamical system could account for the neural activity. To this end, we first produced a de-noised low-dimensional neural state ($X$) by reducing the dimensionality of the neuronal responses to 10 using PCA. Second, we numerically differentiated $X$ to estimate the derivative, $\dot{X}$. Next, we used regression to fit a linear model, predicting the derivative of the neuronal state from the current state: $\dot{X} = MX$. Finally, we computed the fraction of variance explained (FVE) by this model:

$$FVE = 1 - \left\| \dot{X} - MX \right\|_{fro}^2 / \left\| \dot{X} - \langle \dot{X} \rangle \right\|_{fro}^2 \tag{1}$$

$M$ was constrained to be skew-symmetric ($M_{skew}$) unless otherwise specified; $\langle \cdot \rangle$ indicates the mean of a matrix across samples, but not across dimensions; and $\|\cdot\|_{fro}$ indicates the Frobenius norm of a matrix. Unless otherwise specified, analysis of reaching data from each monkey was fourfold cross-validated, whereas analysis of grasp data was 5-fold cross-validated.

## Control comparisons between arm and hand data

We performed several controls comparing arm and hand data to ensure that our results were not an artifact of trivial differences in the data or pre-processing steps.

First, we considered whether alignment of the data to different events might impact results. For the arm data, we aligned each trial to target onset and movement onset (*Figure 1—figure supplement 2A*). For the hand data, we aligned each trial to presentation of the object, movement onset, and the time at which the hand reached maximum aperture during grasp (*Figure 1—figure supplement 2B*). Linear dynamics were strongest (although still very weak) when neuronal responses were aligned to maximum aperture, so this alignment is reported throughout the main text.

Second, we assessed whether rotations might be obscured due to differences in firing rates in the hand vs. arm responses. To this end, we compared peak firing rates for trial-averaged data from the arm and hand after pre-processing (excluding normalization) to directly contrast the inputs to the jPCA analysis given the two effectors/tasks (*Figure 1—figure supplement 2C*). Peak firing rates were actually higher for the hand than the arm, eliminating the possibility that our failure to observe dynamics during grasp was a consequence of weak responses. We also verified that differences in dynamics could not be attributed to differences in the degree to which neurons were modulated in the two tasks. To this end, we computed the modulation range (90th percentile firing – 10th percentile firing) and found that modulation was similar in reach and grasp (*Figure 1—figure supplement 2D*).

Third, we assessed whether differences in the sample size might contribute to differences in variance explained (*Figure 1—figure supplement 2E*). To this end, we took five random samples of 55 neurons from the reaching data set – chosen to match the minimum number of neurons in the grasping data – and computed the cross-validated fraction of variance explained by the rotational dynamics. The smaller samples yielded identical fits as the full sample.

Fourth, we assessed whether the low variance explained by linear dynamics in the hand might be due to poor sampling of the joint motion space (*Figure 1—figure supplement 2G*). To this end, we computed FVE for only rightward reaches, and found that the variance explained for all directions versus only rightward reaches were comparable. Therefore, we expect that our sampling of hand motions would not affect our ability to observe linear dynamics.

Fifth, we considered whether our smoothing kernel might impact results (*Figure 1—figure supplement 2H*). We compared the FVE for the optimal linear dynamical system across various smooth kernels – from 5 to 40 ms – and found that the difference between hand and arm dynamics remains substantial regardless of kernel width.

Finally, since our analyses involve averaging across trials, we assessed whether trial-to-trial variability was different for reach and grasp. To this end, we computed for each neuron the coefficient of variation (CV) of spike counts over 100 ms bins around movement onset. We found the trial-to-trial variability to be stable over the trial and similar for reach and grasp (*Figure 1—figure supplement 2I*).

## Decoding

### Preprocessing

For decoding, we preprocessed the neural data using one of two methods: smoothing with a Gaussian kernel ($\sigma$ = 20 ms) or latent factor analysis via dynamical systems (LFADS) (*Pandarinath et al., 2018*). LFADS is a generative model that assumes that observed spiking responses arise from an underlying dynamical system and estimates that system using deep learning. We used the same number of neurons in the reaching and grasping analyses to train the LFADS models and fixed the number of factors in all models to 30, at which performance of both reach and grasp models had levelled off (*Figure 2—figure supplement 1C*). We allowed two continuous controllers while training the model, which could potentially capture the influence of external inputs on dynamics (*Pandarinath et al., 2018*), since these had significant positive influence on decoding performance (*Figure 2—figure supplement 1D*). Hyper-parameter tuning was performed as previously described (*Keshtkaran and Pandarinath, 2019*).

## Neural reconstruction

To compare our ability to reconstruct single-trial responses using Gaussian smoothing and LFADS, we first computed peri-event time histograms (PETHs) within condition using all training trials (excluding one test trial). We then computed the correlation between the firing rates of each test trial (smoothed with a Gaussian kernel or reconstructed with LFADS) with the PETH of the corresponding condition averaged across the training trails (*Figure 2—figure supplement 1A*). We repeated this procedure with a different trial left out for each condition. We report the difference in correlation coefficient obtained after LFADS processing and Gaussian smoothing (*Figure 2—figure supplement 1B*).

## Kalman filter

To predict hand and arm kinematics, we applied the Kalman filter (*Kalman, 1960*), commonly used for kinematic decoding (*Menz et al., 2015*; *Okorokova et al., 2020*; *Wu et al., 2004*). In this approach, kinematic dynamics can be described by a linear relationship between past and future states:

$$x_t = Ax_{t-1} + v_t \tag{3}$$

where $x_t$ is a vector of joint angles at time $t$, $A$ is a state transition matrix, and $v_t$ is a vector of random numbers drawn from a Gaussian distribution with zero mean and covariance matrix $V$. The kinematics $x_t$ can be also explained in terms of the observed neural activity $z_{t-\Delta}$:

$$x_t = Bz_{t-\Delta} + w_t \tag{4}$$

Here, $z_{t-\Delta}$ is a vector of instantaneous firing rates across a population of M1 neurons at time $t-\Delta$, preprocessed either with Gaussian kernel or LFADS, $B$ is an observation model matrix, and $w_t$ is a random vector drawn from a Gaussian distribution with zero mean and covariance matrix $W$. We tested multiple values of the latency, $\Delta$, and report decoders using the latency that maximized decoder accuracy (150 ms).

We estimated the matrices $A$, $B$, $V$, $W$ using linear regression on each training set, and then used those estimates in the Kalman filter update algorithm to infer the kinematics of each corresponding test set (see *Faragher, 2012*; *Okorokova et al., 2015* for details). Briefly, at each time $t$, kinematics were first predicted using the state transition equation (3), then updated with observation information from equation (4). Update of the kinematic prediction was achieved by a weighted average of the two estimates from (3) and (4): the weight of each estimate was inversely proportional to its uncertainty (determined in part by $V$ and $W$ for the estimates based on $x_{t-1}$ and $z_{t-\Delta}$, respectively), which changed as a function of time and was thus recomputed for every time step.

To assess decoding performance, we performed 10-fold cross-validation in which we trained the parameters of the filter on a randomly selected 90% of the trials and tested the model using the remaining 10% of trials. Importantly, we trained separate Kalman filters for the two types of neural preprocessing techniques (Gaussian smoothing and LFADS) and then compared their performance on the same trials. Performance was quantified using the coefficient of determination ($R^2$) for the held-out trials across test sets.

## Tangling

We computed tangling of the neural population data (reduced to 20 dimensions by PCA) using a published method (*Russo et al., 2018*). In brief, the tangling metric estimates the extent to which neural population trajectories are inconsistent with what would be expected if they were governed by an autonomous dynamical system, with smaller values indicating consistency with such dynamical structure. Specifically, tangling measures the degree to which similar neural states, either during different movements or at different times for the same movement, are associated with different derivatives. This is done by finding, for each neural state (indexed by $t$), the maximum value of the tangling metric $Q(t)$ across all other neural states (indexed by $t'$):

$$Q(t) = \max_{t'} \frac{\|\dot{x}_t - \dot{x}_{t'}\|^2}{\|x_t - x_{t'}\|^2 + \varepsilon} \tag{2}$$

Here, $x_t$ is the neural state at time $t$ (a 20-dimensional vector containing the neural responses at that time), $\dot{x}_t$ is the temporal derivative of the neural state (estimated numerically), and $\|\cdot\|$ is the Euclidean norm, while $\varepsilon$ is a small constant added for robustness to noise (*Russo et al., 2018*). This analysis is not constrained to work solely for neural data; indeed, we also apply this same analysis to trajectories of joint angular kinematics to compare their tangling to that of neural trajectories.

The neural data were pre-processed using the same alignment, trial averaging, smoothing, and normalization methods described above. Joint angles were collected for both hand and arm data. For this analysis, joint angle velocity and acceleration were computed (six total dimensions for arm, 90 dimensions for hand). For reaching, we analyzed the epoch from 200 ms before to 100 ms after movement onset. For grasping, we analyzed the epoch starting 200 ms before to 100 ms after maximum aperture. Neuronal responses were binned in 10 ms bins to match the sampling rate of the kinematics.

The tangling metric is partially dependent on the dimensionality of the underlying data. To eliminate the possibility that our results were a trivial consequence of selecting a particular number of principal components, we tested tangling at different dimensionalities and selected the dimensionality at which Q had largely leveled off for both the population neural activity and kinematics (*Figure 3—figure supplement 1*). Namely, we report results using six principal components (the maximum) for reach kinematics and their associated neural responses, and using 20 for kinematics and neuronal responses during grasp.

## Dimensionality of the neuronal response

One possibility is that our failure to observe autonomous dynamics during grasp stems from a failure to properly characterize the neural manifold, which in principle could be much higher dimensional for grasp than it is for reach. However, the first D dimensions of a manifold can be reliably estimated from fewer than 2*D projections if two conditions hold: the eigenvalue spectrum is not flat, and the samples approximate random projections of the underlying manifold (*Halko et al., 2011*). The scree plot shows that the first condition is met (*Figure 1—figure supplement 1F*). To evaluate the second condition and determine whether neurons are random projections of the low-dimensional manifold, we applied a Gine-Ajne test (*Prentice, 1978*) to the first 5, 10, and 20 PCs. We found that the null hypothesis of spherical uniformity was not rejected (p>0.5 for all dimensionalities and data sets). While we cannot rule out that the possibility that there exists a small, unrecorded fraction of neurons that span a disjoint manifold subspace from that we measured, the failure to reject spherical uniformity provides evidence that these neurons approximate random projections. To further examine the possibility that dynamics occupy a space that we were unable to resolve with our neuronal sample, we implemented LFADS with a different number of latent factors. We found that, to the extent that decoding performance improved with additional latent factors, it levelled off at ~10 factors (*Figure 2—figure supplement 1*). If the dynamics were distributed over a high-dimensional manifold, we might expect that performance would increase slowly with the number of latent factors over the entire range afforded by the sample size. This was not the case.

Yet another possibility we considered is that the neuronal manifold beyond the first few dimensions reflects noise, which would preclude the identification of dynamics embedded in higher order dimensions. To examine this possibility, we assessed our ability to relate the monkeys' behavior during the grasp task to the neural data over subsets of dimensions. First, we found that the ability to classify objects based on the population response projected on progressively smaller subspaces – removing high-variance principal components first – remained above chance even after dozens of PCs were removed. This suggests that behaviorally relevant neuronal activity was distributed over many dimensions, and that this signal clearly rose above the noise (*Figure 3—figure supplement 2A*). For this analysis, we used multiclass linear discriminant analysis based on population responses evoked over a 150 ms window before object contact. Second, we found that the ability to decode kinematics based on the population response projected on progressively smaller subspaces remained above chance after removal of many PCs, consistent with the classification analysis (*Figure 3—figure supplement 2B*). For this analysis, we used population responses over an 800 ms window centered on maximum aperture for reaching and movement onset for grasping. Thus, high-order PCs do not simply reflect noise but rather comprise behaviorally relevant signals.

In summary, then, our sample size is sufficient, in principle, to recover dynamics embedded in a high-dimensional manifold. The weak dynamics in the grasping response that we did recover occupy a low-dimensional manifold, and we were able to resolve the population response for the grasping behavior across a large number of dimensions (40+ principal components).

## Statistics

For most of analyses, sample sizes were large and data were distributed approximately normally so we used two-sided t-test. However, for some analyses, the data were right-skewed and the sample size was small, so we used non-parametric tests, either the Wilcoxon signed rank test or the Mann-Whitney-Wilcoxon test depending on whether the samples were matched (for example, comparison of same kinematic DoFs reconstructed with either Gaussian smoothing or LFADS) or not (for example, comparison of kinematic DoFs reconstruction from different datasets).

## Acknowledgements

We thank Sangwook A Lee, Gregg A Tabot and Alexander T Rajan for help with data collection, as well as Mohammad Reza Keshtkaran and Chethan Pandarinath for help with the LFADS implementation. This work was supported by NINDS grants NS082865, NS101325, NS096952, NS045853, and NS111982.

## Additional information

### Funding

| Funder | Grant reference number | Author |
|---|---|---|
| National Institute of Neurological Disorders and Stroke | NS082865 | Nicholas G Hatsopoulos Sliman J Bensmaia |
| National Institute of Neurological Disorders and Stroke | NS096952 | Aneesha K Suresh |
| National Institute of Neurological Disorders and Stroke | NS045853 | Nicholas G Hatsopoulos |
| National Institute of Neurological Disorders and Stroke | NS111982 | Nicholas G Hatsopoulos |
| National Institute of Neurological Disorders and Stroke | NS101325 | Sliman J Bensmaia |

The funders had no role in study design, data collection and interpretation, or the decision to submit the work for publication.

### Author contributions

Aneesha K Suresh, Conceptualization, Data curation, Formal analysis, Writing - original draft, Writing - review and editing; James M Goodman, Data curation, Formal analysis, Investigation, Writing - original draft, Writing - review and editing; Elizaveta V Okorokova, Formal analysis, Writing - review and editing; Matthew Kaufman, Formal analysis, Supervision, Writing - review and editing; Nicholas G Hatsopoulos, Conceptualization, Funding acquisition, Investigation, Project administration, Writing - review and editing; Sliman J Bensmaia, Conceptualization, Supervision, Funding acquisition, Writing - original draft, Project administration, Writing - review and editing

### Author ORCIDs

Aneesha K Suresh (iD) https://orcid.org/0000-0002-1014-9541
James M Goodman (iD) https://orcid.org/0000-0001-6055-0600
Elizaveta V Okorokova (iD) https://orcid.org/0000-0002-2719-2706
Sliman J Bensmaia (iD) https://orcid.org/0000-0003-4039-9135

## Ethics

Animal experimentation: All surgical, behavioral, and experimental procedures conformed to the guidelines of the National Institutes of Health and were approved by the University of Chicago Institutional Animal Care and Use Committee (#72042).

## Decision letter and Author response

Decision letter https://doi.org/10.7554/eLife.58848.sa1
Author response https://doi.org/10.7554/eLife.58848.sa2

# Additional files

## Supplementary files

• Supplementary file 1. Datasets and related analyses and figures.

## Data availability

The data that support the findings of this study have been deposited in Dryad, accessible at https://doi.org/10.5061/dryad.xsj3tx9cm.

The following dataset was generated:

| Author(s) | Year | Dataset title | Dataset URL | Database and Identifier |
|---|---|---|---|---|
| Suresh AK, Goodman JM, Okorokova EV, Kaufman MT, Hatsopoulos NG, Bensmaia SJ | 2020 | Neural population dynamics in motor cortex are different for reach and grasp | http://dx.doi.org/10.5061/dryad.xsj3tx9cm | Dryad Digital Repository, 10.5061/dryad.xsj3tx9cm |

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
