## [Decision Letter]

**Acceptance summary:**

The authors present a short report demonstrating the difference in neural dynamics between grasping and reaching behaviors. This work is broadly interesting to those in the field of motor control and leverages cutting-edge techniques to elucidate neural dynamics associated with the two aforementioned motor behaviors. We are enthusiastic about the suitability of this publication in *eLife*.

**Decision letter after peer review:**

Thank you for submitting your article "Neural population dynamics in motor cortex are different for reach and grasp" for consideration by *eLife*. Your article has been reviewed by three peer reviewers, including Samantha R Santacruz as the Reviewing Editor and Reviewer #1, and the evaluation has been overseen by Richard Ivry as the Senior Editor. The following individual involved in review of your submission has agreed to reveal their identity: Marco Capogrosso (Reviewer #2).

The reviewers have discussed the reviews with one another and the Reviewing Editor has drafted this decision to help you prepare a revised submission.

Summary:

The authors present a cohesive and elegant short report demonstrating the difference in neural dynamics between grasping and reaching behaviors. The reviewers are enthusiastic about this work and agree that this study is of great interest to the field since increasingly the motor cortex is modelled as a system with strong dynamical properties. However, they find that the manuscript would be greatly strengthened by clarifications in the analyses, statistics, and animals utilized. The manuscript is suitable for publication in *eLife* subject to the revisions detailed below.

Essential revisions:

1) Analysis to convince the reader that motor cortical neural activity analyzed is as grasp-modulated as much as it is reach-modulated, which would control for the possibility that neural activity is just more reach-modulated so looks like reach conditions have stronger dynamics. Are the percentage of neurons modulated by the task same in grasp vs. reach (in the PSTH that you analyze)? Since these dynamics questions reflect how well changes (modulation) in firing rate are predicted, it would be important to know that the amount of modulation is comparable. Further, please clarify the point articulated in paragraph three of subsection “Control comparisons between arm and hand data”. Since firing rates are normalized before jPCA, why is analyzing the peak firing rates without normalization a valid way to "directly contrast the inputs to the jPCA analysis"?

2) Analysis to show that reach and grasp PSTHs are equally representative of individual trials, which would control for the possibility that grasp activity is just more variable trial-to-trial so analyzing the PSTH isn't representative of true dynamics. How reliable is the trial-to-trial neural activity for reach vs. grasp? Ensuring that the PSTH is equally reflective of trial activity is important for fairly comparing these two conditions.

3) Please report R2 for the neural reconstruction with LFADS for reach vs. grasp in Figure 2. This value would indicate whether using a non-linear dynamics model (LFADS) can accurately predict neural activity even in the case of grasp, which is important to do prior to any of the kinematic decoding.

4) Clarify the number of animals used for each analysis. It is difficult to understand from the results and reported figures how many animals were used and for which analysis. We suggest using a table to report this information in an accessible format. When performing statistics with data combined across animals, we also suggest using a linear mixed effect model with "animal" as a random effect.

[Editors' note: further revisions were suggested prior to acceptance, as described below.]

Thank you for resubmitting your article "Neural population dynamics in motor cortex are different for reach and grasp" for consideration by *eLife*. Your revised article has been reviewed by two of the original peer reviewers, and the evaluation has been overseen by a Reviewing Editor and Richard Ivry as the Senior Editor.

The reviewers have discussed the reviews with one another and the Reviewing Editor has drafted this decision to help you prepare a revised submission.

Summary:

The authors present a short report demonstrating the difference in neural dynamics between grasping and reaching behaviors. The reviewers remain enthusiastic about this work and the overall interest that it will have to the field, but there remain outstanding concerns regarding the statistics and interpretation of results. Below you will find more detailed comments. The manuscript is suitable for publication in eLife*eLife* if the points detailed below can be addressed.

– Pertaining to Essential Revisions #3: The authors now report an R2 for the NEURAL reconstruction using LFADS for reaching and grasping. The authors have placed this result in the Materials and methods section, rather than in the Results section. Secondly and more importantly, this result is: "The average correlation between measured and reconstructed firing rates was 0.44 +/- 0.022 and 0.48 +/- 0.021 for single trials and 0.73 +/- 0.03 and 0.76 +/- 0.011 when averaged within condition, for reach and grasp respectively". This suggests that both reach and grasp NEURAL activity are equally explained by LFADS. This result appears to go against the main message of their paper (which to this point has been that there are no discernible dynamics in grasping, but there are in reaching). We would like to see this result reported in the Results section before Figure 2, and would like to see the message of the paper reflect this result (maybe something along the lines of "Grasping dynamics are high dimensional, non-linear, and can't be used for decoding with a linear decoder whereas reaching dynamics are low dimensional, linear, and can be used for decoding").

– The R2 result from above also seems to contradict the tangling results in Figure 3 (that Q-M1/Q-kinematics is higher for grasping than reaching). However upon further inspection of Figure 3, it seems like the reaching and grasping Q-kinematics are quite different (mean Q-kinematics seems to be about ~1x10^4^ for reaching, ~0.3x10^4^ for grasping), whereas it looks like the Q-motor cortex may be similar for both reaching and grasping. Perhaps the kinematics themselves may be driving the significant differences in the Q-ratio while the Q-motor cortex values may be comparable (which would be more consistent with the above result for approx equal R2 from LFADS)? This should be addresses in the revision.

– Pertaining to essential revisions #2: We appreciate the inclusion of panel I in Figure 1—figure supplement 2 to address this point. The main point of this question was to assess whether trial-to-trial variability affected the estimate of the PSTH and thus the ability of a linear model to capture dynamics from the PSTH. Displaying the coefficient of variation as a bar graph collapses over all temporal differences in trial-to-trial variability. For example, it is consistent within this bar plot that trial-to-trial activity is approx. uniform across the reaching behavior epoch, but for grasp is low at the beginning of the trial then high at the end of the trial for example. This hypothetical difference would make it so that the grasping PSTH is consistent at the beginning and noisy at the end, and could explain why it is harder to estimate grasping PSTH with linear dynamics. If this is the case, it may be that reach and grasp neural dynamics are not very different, just that grasp behavior tends to be more variable so the PSTH is not reflective of the true dynamics that may be ongoing during grasp. Another way to address this concern would be to report R2 of neural activity estimated from fitting dynamics on single trials and showing the same differences as in Figure 1. This gets around the issue of trial-averaging and potential trial-to-trial variability differences. We ask that the authors report this R2 value.

– There remain some overall concerns with the statistics performed. When performing statistics, data points from different subjects cannot be pooled together. This is because performing tests on pooled data violates the assumption of iid samples because part of the variance in the samples is explained by the fact that data some of the samples are from one animal and some from the other (intra-animal vs inter-animal). In this case manuscript, the authors are comparing 2 monkeys against 2 different monkeys, and everything is pooled together. We ask the authors to clarify and justify their methodology.

---

## [Author Response]

Essential revisions:1) Analysis to convince the reader that motor cortical neural activity analyzed is as grasp-modulated as much as it is reach-modulated, which would control for the possibility that neural activity is just more reach-modulated so looks like reach conditions have stronger dynamics. Are the percentage of neurons modulated by the task same in grasp vs. reach (in the PSTH that you analyze)? Since these dynamics questions reflect how well changes (modulation) in firing rate are predicted, it would be important to know that the amount of modulation is comparable. Further, please clarify the point articulated in paragraph three of subsection “Control comparisons between arm and hand data”. Since firing rates are normalized before jPCA, why is analyzing the peak firing rates without normalization a valid way to "directly contrast the inputs to the jPCA analysis"?

Thank you for this comment, we have addressed this point in Figure 1—figure supplement 2, Panel D, and in the relevant passage of the text. The goal was to directly address the reviewer’s question, namely whether firing rates are similar across neuronal populations and tasks. The modulation depths were similar for reach and grasp responses.

2) Analysis to show that reach and grasp PSTHs are equally representative of individual trials, which would control for the possibility that grasp activity is just more variable trial-to-trial so analyzing the PSTH isn't representative of true dynamics. How reliable is the trial-to-trial neural activity for reach vs. grasp? Ensuring that the PSTH is equally reflective of trial activity is important for fairly comparing these two conditions.

We assessed trial-to-trial variability by computing the coefficient of variation of spike counts over a 500-ms window centred on movement onset. The results of this analysis are shown in Figure 1—figure supplement 2, Panel I. Trial-to-trial variability was similar for reach and grasp.

3) Please report R2 for the neural reconstruction with LFADS for reach vs. grasp in Figure 2. This value would indicate whether using a non-linear dynamics model (LFADS) can accurately predict neural activity even in the case of grasp, which is important to do prior to any of the kinematic decoding.

We now report average correlations between firing rates estimated with Gaussian smoothing and those estimated with LFADS. We find that the correlations between the two are similar for reach and grasp in both condition-averaged responses and on a trial-by-trial basis (now reported in the Materials and methods section). This result indicates that the model captured similar amount of variance in reach and grasp datasets, yet this variance was more informative about kinematics for reach than for grasp, as evidenced by the decoding analysis.

4) Clarify the number of animals used for each analysis. It is difficult to understand from the results and reported figures how many animals were used and for which analysis. We suggest using a table to report this information in an accessible format. When performing statistics with data combined across animals, we also suggest using a linear mixed effect model with "animal" as a random effect.

We have added a table to provide the requested information. We used different monkeys in the reach and grasp tasks, so unfortunately our data do not admit the suggested repeated-measures analysis design.

[Editors' note: further revisions were suggested prior to acceptance, as described below.]

Revisions for this paper:– Pertaining to Essential Revisions #3: The authors now report an R2 for the NEURAL reconstruction using LFADS for reaching and grasping. The authors have placed this result in the Materials and methods section, rather than in the Results section. Secondly and more importantly, this result is: "The average correlation between measured and reconstructed firing rates was 0.44 +/- 0.022 and 0.48 +/- 0.021 for single trials and 0.73 +/- 0.03 and 0.76 +/- 0.011 when averaged within condition, for reach and grasp respectively". This suggests that both reach and grasp NEURAL activity are EQUALLY explained by LFADS. This result appears to go against the main message of their paper (which to this point has been that there are no discernible dynamics in grasping, but there are in reaching). We would like to see this result reported in the Results section before Figure 2, and would like to see the message of the paper reflect this result (maybe something along the lines of "Grasping dynamics are high dimensional, non-linear, and can't be used for decoding with a linear decoder whereas reaching dynamics are low dimensional, linear, and can be used for decoding").

We thank the reviewers for this comment. In the first revision, we reported mean correlations between the trial-averaged responses and their smoothed and LFADS-processed counterparts. This was a mistake. What we should have done instead is to compute the correlation between the response obtained on one trial and the smoothed or LFADS-processed response averaged over the other trials. We would then predict that, for reaching, LFADS should yield responses that generalize better because it leverages the latent dynamics to reconstruct the single trial response. This is indeed what we found for the reaching responses and significantly less so for the grasping responses:

“First, as expected, we found that in both datasets, neural reconstruction of single trials improved with LFADS (0.34 and 0.23 correlation improvement in reach and grasp, correspondingly, Figure 2—figure supplement 1 (A, B)). However, neural reconstruction improvement was on average significantly higher for reach than for grasp (t(311) = 7.07, p = 5.11e-12; Figure 2—figure supplement 1).”

– The R2 result from above also seems to contradict the tangling results in Figure 3 (that Q-M1/Q-kinematics is higher for grasping than reaching). However upon further inspection of Figure 3, it seems like the reaching and grasping Q-kinematics are quite different (mean Q-kinematics seems to be about ~1x10^4^ for reaching, ~0.3x10^4^ for grasping), whereas it looks like the Q-motor cortex may be similar for both reaching and grasping. Perhaps the kinematics themselves may be driving the significant differences in the Q-ratio while the Q-motor cortex values may be comparable (which would be more consistent with the above result for approx equal R2 from LFADS)? This should be addresses in the revision.

Now that we have done the analysis properly, the contradiction between the LFADS reconstruction and the tangling analysis is resolved. Regarding the raw tangling values, these depend on task, number of conditions, time binning, smoothing, and other factors, and thus fundamentally constitute a relative measure (Russo et al., 2018). The kinematics are matched for the aforementioned factors and thus make a nice comparison.

– Pertaining to essential revisions #2: We appreciate the inclusion of panel I in Figure 1—figure supplement 2 to address this point. The main point of this question was to assess whether trial-to-trial variability affected the estimate of the PSTH and thus the ability of a linear model to capture dynamics from the PSTH. Displaying the coefficient of variation as a bar graph collapses over all temporal differences in trial-to-trial variability. For example, it is consistent within this bar plot that trial-to-trial activity is approx. uniform across the reaching behavior epoch, but for grasp is low at the beginning of the trial then high at the end of the trial for example. This hypothetical difference would make it so that the grasping PSTH is consistent at the beginning and noisy at the end, and could explain why it is harder to estimate grasping PSTH with linear dynamics. If this is the case, it may be that reach and grasp neural dynamics are not very different, just that grasp behavior tends to be more variable so the PSTH is not reflective of the true dynamics that may be ongoing during grasp. Another way to address this concern would be to report R2 of neural activity estimated from fitting dynamics on single trials and showing the same differences as in Figure 1. This gets around the issue of trial-averaging and potential trial-to-trial variability differences. We ask that the authors report this R2 value.

The linear dynamical analysis cannot be applied to single trials because these are too noisy. Indeed, even the much more constrained jPCA cannot be computed from single trials (see Figures 3B,D in Pandarinath et al., 2018). LFADS, on the other hand, is well suited for this purpose. Accordingly, the analysis described above is in the spirit of what is requested here. To address the specific possibility, raised by the reviewer, that the variability may be distributed differently within the trial for reaching and grasping, we recomputed the CV at different epochs during each trial and found these to be homogeneous over the trial and similar for reaching and grasping (see Figure 1—figure supplement 2).

– There remain some overall concerns with statistics performed. When performing statistics, data points from different subjects cannot be pooled together. This is because performing tests on pooled data violates the assumption of iid samples because part of the variance in the samples is explained by the fact that data some of the samples are from one animal and some from the other (intra-animal vs inter-animal). In this case manuscript, the authors are comparing 2 monkeys against 2 different monkeys, and everything is pooled together. We ask the authors to clarify and justify their methodology.

We agree that it would have been preferable to obtain reaching and grasping data from the same animals. However, the two tasks were used in two different studies with different goals and, unfortunately, different animals. Note, however, that the differences between reach and grasp are very strong and persist even if we compare the least favorable pair of animals.